# Upgrades of a Small Electrostatic Dust Accelerator at the University of Stuttgart

**Yanwei Li** [1,*] **, Marcel Bauer** [1] **, Sebastian Kelz** [2] **, Heiko Strack** [1] **, Jonas Simolka** [1] **, Christian Mazur** [1] **, Maximilian Sommer** [1] **, Anna Mocker** [1] **and Ralf Srama** [1]

[1] Institut für Raumfahrtsysteme, Universität Stuttgart, Pfaffenwaldring 29, 70569 Stuttgart, Germany
[2] Institut für Elektrische und Optische Nachrichtentechnik, Universität Stuttgart, Pfaffenwaldring 47, 70569 Stuttgart, Germany
* Correspondence: li@irs.uni-stuttgart.de; Tel.: +49-711-685-69653

**Abstract:** In this paper, we describe the upgrade of a small electrostatic dust accelerator located at the University of Stuttgart. The newly developed dust source, focusing lens, differential detector and linac stage were successfully installed and tested in the beam line. The input voltage range of the dust source was extended from 0–20 kV to 0–30 kV. A newly developed dust detector with two differential charge sensitive amplifiers is employed to monitor particles with speeds from several m/s to several km/s and with surface charges above 0.028 fC. The post-stage linac provides an additional acceleration ability with a total voltage of up to 120 kV. The entire system of this dust accelerator works without protection gas and without a complex high voltage terminal. The volumes to be pumped down are small and can be quickly evacuated. The new system was used to accelerate micron- and submicron-sized metal particles or coated mineral materials. Improvements in the acceleration system allow for a wider variety of dust materials and new applications.

**Keywords:** dust; electrostatic accelerator; dust source; linac; charge sensitive amplifier

## 1. Introduction

Micron-sized particles acceleration can be achieved using plasma acceleration [1], light gas guns [2], laser pulses [3], as well as electrostatic acceleration. The latter method is the most promising solution to obtain a continuous and stable beam with individual high speed particles. The dust accelerator system normally consists of a dust source, an acceleration stage, dust monitors, a Particle Selection Unit (PSU) and an experiment chamber. The dust particles are charged in the dust source and they acquire kinetic energy when they pass a potential difference. The accelerated dust particles carry surface charges that can be characterized with dust detectors using charge induction methods. Individual dust particles are monitored, selected and recorded with the PSU.

The first electrostatic dust accelerator was reported by Shelton [4] to study impact phenomena of single high speed particles. It consists of a dust source connected to a high voltage generator (100 kV), a dust detector and a test chamber. Similar systems were developed further with the 2 MV Van de Graaff accelerator at the TRW Space Technology Laboratories [5], the 2 MV Van de Graaff accelerator at the University of Stuttgart [6], the 2 MV Van de Graaff accelerator at the University of Kent [7], the 3.75 MV Van de Graaff accelerator at the University of Tokyo [8] and the 3 MV Pelletron accelerator at the University of Colorado (Boulder) [9].

For simplified operation of these large scale accelerator facilities, small dust accelerators with low potential differences are synchronously built to test dust sources and dust samples. The entire system of these small accelerators work without a tank filled with protection gas and without a special high voltage terminal. Hence, the volumes to be pumped down are small and can be quickly evacuated. The small dust accelerators are

very useful to determine the dust source function before mounting them in a big accelerator facility. Small dust accelerators usually work with acceleration voltages of up to 20 kV and they can launch micron-sized particles with speeds up to several km/s.

The low velocity range of such accelerators allows us to study a variety of science cases that are relevant in planetary science. Laboratory research recently provided evidence that dust particles above an airless object can be charged under UV radiation and/or plasmas and levitated with local electric fields [10]. For example, the micron-sized dust particles covering the lunar surface are expected to be electrostatically charged by their interaction with the solar wind, UV and X-ray radiation and the local plasma environment [11]. The charged particles are proposed to be naturally lifted by near-surface electric fields [12]. The data gathered by the LDEX detector onboard NASA's LADEE mission show that there is a permanent, asymmetric dust cloud around the Moon [13]. The density of the dust grains in the cloud varies with annual meteoroid showers. Micron-sized secondary ejecta particles generated by these meteoroid showers and interplanetary dust are considered as the main source of the lunar dust cloud, which were launched from the lunar surface with speeds up to several km/s.

Over the next decades, space agencies and private players have plans for missions (Lunar Gateway, Chang'e-6 and Luna 25, etc.) to send spacecrafts and astronauts to the surface of our nearest neighbour, the Moon. Dust is considered one of the highest-priority challenges for these future robotic or human explorations [14,15], which should be studied in depth with in situ detectors to develop an environmental model for future lunar exploration. The velocities of dust particles launched by small accelerators are sufficient to calibrate the dust sensors used for the detection of secondary ejecta and levitated dust, which provides new application opportunities for small dust accelerators. In this article, we present our new upgrades of our 20 kV dust accelerator and related parameters of the accelerated dust particles.

## 2. Materials and Methods

A compact electrostatic dust accelerator mainly consists of one high voltage power supply, one dust source, one focusing lens, dust detectors, target chambers and a high vacuum system. Our previous small accelerator is shown in Figure 1. A schematic diagram of our upgraded small dust accelerator with the PSU with 3 additional dust detectors and deflection stage is given in Figure 2. The test setup of the linac stage consists of one dust source, one focussing system, one differential dust detector (2 CSA channels), the linac stage and one single end dust detector after the linac (see Figure 3).

The dust source is the heart of an electrostatic dust accelerator first developed by reference [4]. The dust source in small accelerators serves two purposes: charging dust particles and accelerating them to final speeds. Dust particles are filled into the reservoir during assembly. A pulser unit connected to the reservoir applies a time-varying electric field between the needle and the reservoir. Dust particles are charged, lifted and pushed through a small hole out of the reservoir. A grounded extraction plate further accelerates charged particles to their final speeds. A beam collimator system with grounded metal rings with a diameter of 0.8 mm leads particles into the accelerator beam line. The entire accelerator employs a high-vacuum system to avoid sparking caused by the high electric field strength in the dust source.

Dust particles are fired from the dust source using a potential difference of the needle to ground. After exiting the collimating system, the dust particles are focused with an electrostatic lens. The accelerated dust particles are characterized while passing the dust detectors. The upgraded beam line provides two test locations: (1) Test chamber 1 is close to the dust source and is suitable for high-frequency particle experiments, especially for particles with speeds around tens m/s; (2) Test chamber 2 can be used together with the PSU (and/or linac) for experiments with particles in specific selection windows with higher speeds.

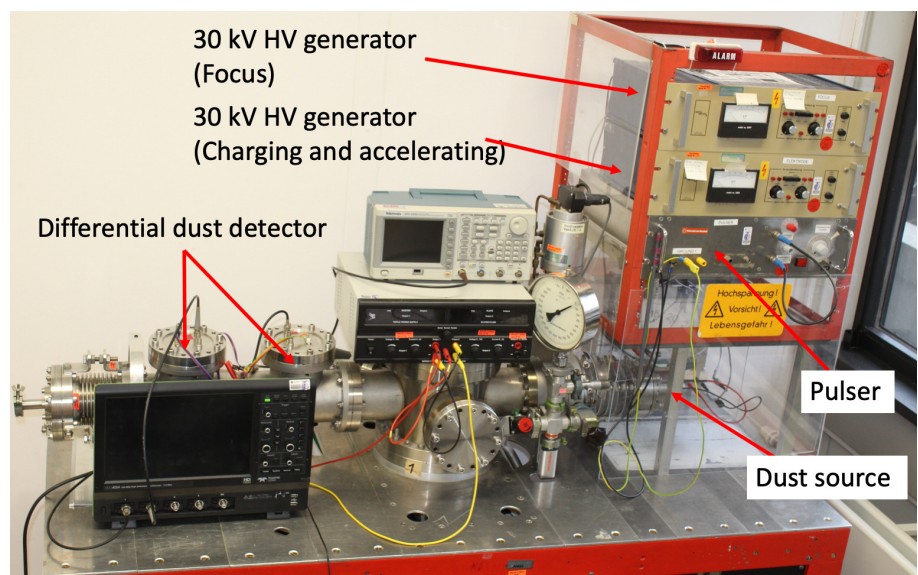

**Figure 1.** Small dust accelerator with short beam line. The high voltage generator (Brandenburg 2807 Alpha Serial II) produces an acceleration voltage up to 30 kV. The dust source is operated in air condition and no protection gas (such as SF6) is required. The beam line itself is evacuated to a vacuum of better than $10^{-6}$ mbar using a turbo molecular pump.

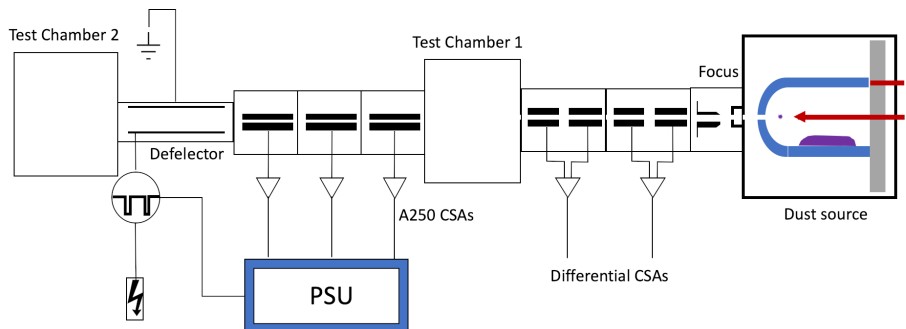

**Figure 2.** Schematic of the entire system of the upgraded small dust accelerator.

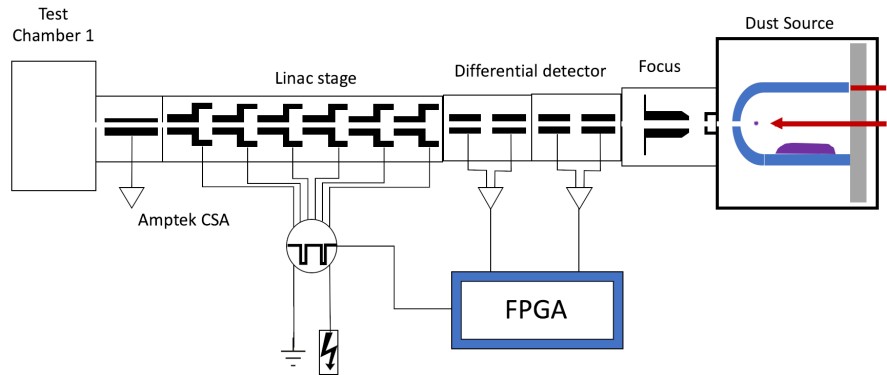

**Figure 3.** Schematic diagram of the entire accelerator with post-stage linac.

A comparison of our previous small accelerator and the upgraded setup is shown in Table 1 and will be described in detail in the following sections.

**Table 1.** Comparison of previous small accelerator and upgraded setup.

|  | Previous Setup | New Setup |
|---|---|---|
| Total length | 1.5 m | Up to 4.5 m |
| Focusing system | Single cylinder | Einzel lens |
| Acceleration voltage | Up to 20 kV | Up to 30 kV (+120 kV) |
| Particle speed detection range | Above 100 m/s | 10 m/s to 10 km/s |
| Test chamber | ⌀ 100 mm | ⌀ 100 mm and ⌀ 300 mm |
| Operation mode | Continuous mode | Continuous and single mode |

The velocity obtained by a particle in an electrostatic field depends on its charge-to-mass ratio and acceleration voltage. If a particle of mass $m$ and surface charge $q$ is accelerated through potential $U_{acc}$, the speed $v$ is obtained from the energy law by:

$$v = \sqrt{2 \cdot \frac{q}{m} \cdot U_{acc}} \tag{1}$$

*2.1. Dust Source Upgrade*

The charging of dust particles is crucial for electrostatic acceleration. Particles may be charged with UV irradiation [16], electron and ion beams, or contact with a charged surface. The latter method, which results in an adequate particle charge-to-mass ratio ($q/m$), was utilized to develop the dust source in the electrostatic accelerator. The newly upgraded dust source is based on our previous design described in [17], which mainly consists of one vacuum housing, one dust reservoir, one needle electrode and one collimating system (see Figure 4). The dust reservoir has a cylindrical shape with a length of 25 mm and a diameter of 12 mm. The 1 mm tungsten needle is very sharp and has a tip diameter in the range of 1–4 μm, which is centered in the cylinder axis of the dust reservoir and aligned to the accelerator beam line. The dust reservoir itself lies on the same electric potential as the needle and is pulsed down frequently to blow up the dust powder. Conductive dust particles are filled into the reservoir. The electric field will induce a charge on the surfaces of single particles and lift them up. Whenever these levitated particles hit the needle tip, they obtain higher electric charges and are ejected. The accelerated dust particles have to pass through a small hole (0.8 mm in diameter) in the extraction plate and after a collimation system they enter the beam line.

Conductive dust particles can obtain surface charges by contact with highly charged surfaces [16]. The surface charge of a spherical particle with radius $r$ under vacuum permittivity $\epsilon_0$ and surface potential $\Phi$ is given by:

$$q = \Phi \cdot 4\pi\epsilon_0 r \tag{2}$$

All particles obtain their final speeds from the dust source in a dust accelerator, shown in Figure 1. For a dust particle of a given radius and density, its final speed is only dependent on the high voltage added in the dust source. As a result, the high-voltage interface used in the dust source has been modified and redesigned. Our new design allows input voltages up to 40 kV through two ceramic feedthroughs (CeramTec) (see the right side of Figure 4). Furthermore, the mechanical designs of the dust reservoir and isolator are simplified to reduce the number of assembled parts and optimise the evacuation by the vacuum pump. The simpler and lighter structure of the new dust source allows an easier handling and a faster dust refilling. The parameters of our upgraded dust source are summarized in Table 2.

**Table 2.** Parameters of the upgraded dust source.

| Parameter | Description | Value |
|---|---|---|
| Size | Length × Diameter<br>Reservoir<br>Distance (Needle-extraction hole) | 127 mm × 127 mm<br>$\phi$ = 12 mm, length = 25 mm<br>2.5–4 mm |
| Material | Chassis<br>Insulators<br>Needle<br>HV feedthrough (40 kV) | Stainless steel<br>PEEK<br>Tungsten<br>Alumina ceramic |
| Voltage | Needle<br>Reservoir | 0–30 kV, fixed<br>same as needle, pulsed |
| Pulses | Duration<br>Repetition | 1–255 ms, adjustable<br>1–255 ms, adjustable |

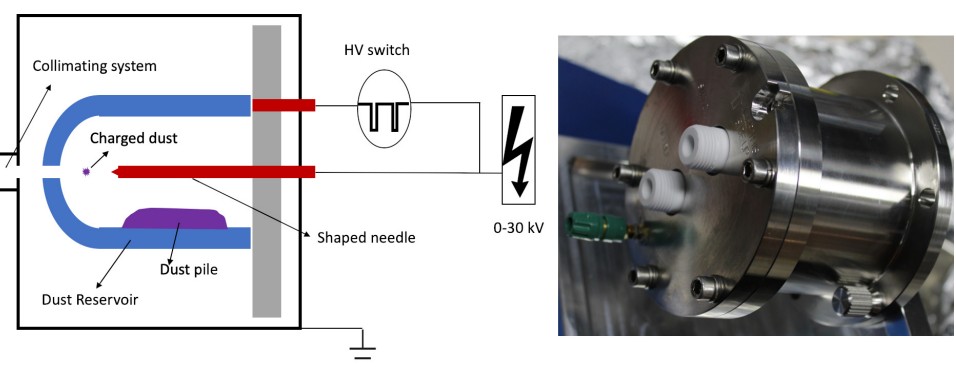

**Figure 4.** Schematic diagram and realistic picture of the new dust source. Dust particles are charged by contact with the reservoir surface. Charged particles will be levitated by the electrostatic force. Particles close to the exiting hole and the needle will be fired out. The potential drops to ground (0 V) on the extraction plate.

## 2.2. Focusing System Upgrade

In order to collimate dust particles, a focusing system located behind the dust source is needed. Charged dust particles can be focused using electric fields (electrostatic force) or magnetic fields (Lorentz force). We compared two electrostatic lens systems (a single-cylinder system and an einzel lens system, see Figure 5) in the small accelerator with an acceleration voltage of 15 kV. The shape of the single-cylinder focusing system is similar to the one used in our 2 MV dust accelerator facility [6]. The particularly shaped metal cylinder with an inner diameter of 8 mm is directly behind the dust source. The einzel lens consists of three separated cylindrical electrodes. The central electrode is connected to high voltages to generate lens effects and the other two electrodes are grounded.

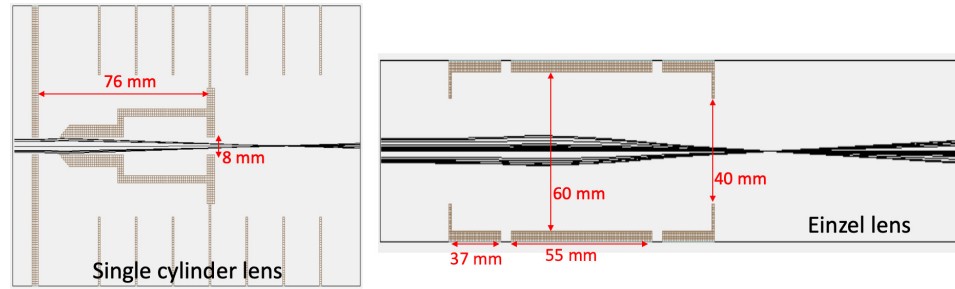

**Figure 5.** Schematic diagram of focal electrodes (**left**: single cylinder system; **right**: einzel lens system) and simulations of the focusing effect in SIMION. The potentials of the electrodes were altered to show the focusing effect.

The relation between the focal length and electrode potential is shown in Table 3. The focal length is calculated between the dust source and the focal point. The focal length is affected by the electrode geometry and focusing and acceleration voltages. Based on our simulation results and operation experiences, for a given acceleration voltage, the focal point of the dust beam is altered by varying the potential of the focusing electrode according to the experimental requirements.

**Table 3.** Lens effect of focusing system based on SIMION simulations.

| Lens | Focusing Voltage | Acceleration Voltage | Focal Length |
|------|------------------|----------------------|--------------|
| Einzel lens | 10 kV | 15 kV | beyond beam (5 m) |
| | 10.745 kV | 15 kV | 3000 mm |
| | 10.94 kV | 15 kV | 1500 mm |
| | 11 kV | 15 kV | 1327 mm |
| | 12 kV | 15 kV | 467 mm |
| Single-cylinder | 10 kV | 15 kV | beyond beam (5 m) |
| | 11 kV | 15 kV | beyond beam (5 m) |
| | 11.15 kV | 15 kV | 3000 mm |
| | 11.18 kV | 15 kV | 1500 mm |
| | 12 kV | 15 kV | 250 mm |

### 2.3. Dust Detector Upgrade

The dust detector is needed to monitor the particles launched by the dust source. It is based on charge induction and enables the non-contact measurement of individual dust particles down to the sub-micron range. Positively charged dust particles attract electrons and induce electron movement in metal electrodes. A charge sensitive amplifier (CSA) connected to the electrode transforms the induced charge to an equivalent voltage, which is evaluated using the detection system. In order to measure hypervelocity dust particles with surface charges down to 0.1 fC, Srama [18] developed a low-noise dust detector. The dust detector employed a copper cylinder electrode with a diameter of 9 mm and a length (*L*) of 200 mm. The particle velocity (*v*) is determined by the flight time (*T*) through the electrode (Equation (3)).

$$v = \frac{L}{T} \tag{3}$$

Such a dust detector has a very high accuracy in the measurement of low- and high-speed particles up to 200 km/s. However, the integrated Amptek A250F/NF CSA in the design has a lower cutoff frequency of 2 kHz, which restricts its application in the measurement of dust particles with speeds below 100 m/s launched by small accelerators. In contrast, our upgraded dust detector consists of four short cylindrical tubes each 5 cm in length and 0.9 cm in inner diameter (see Figure 6). The electrodes are surrounded by two cylindrical Faraday caps to restrain any interference of environmental charges or electromagnetic fields. The electrodes are shielded from each other by grounded grids and insulated by Polytetrafluorethylen (PTFE) supports. The connection of two sensor electrodes to one single differential CSA further reduced the influence of common mode interference. The screening by the shielding cylinders is improved by attaching larger meshes. The electrodes and CSAs are mounted in a 100 mm vacuum pipe with standard CF 100 flange interfaces.

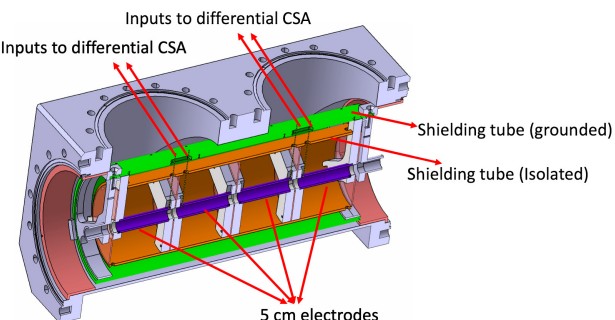

**Figure 6.** Cross section of the differential detector. The detector consists of two differential CSAs, which are not shown here.

The design of the differential CSA is based on an improved version of the design presented in reference [19]. While traditional CSAs evaluate the influenced charge of a single electrode with reference to ground, the detector presented in this paper contains two electrodes and evaluates the charge difference between both electrodes. This results in a bipolar output pulse as given in Figure 7, which in subsequent designs will be filtered with a matched-filter system. The amplifier Application Specific Integrated Circuit (ASIC) is designed in a 350 nm technology. It has a 3 dB frequency range from 4 Hz to 1 MHz and therefore is able to accurately reproduce the influenced charges even of very low-speed particles. The low cutoff frequency of 4 Hz is achieved by replacing the feedback resistor $R_f$ typically required for CSAs with a MOSFET in an off-state. This enables the design to reach an effective feedback resistance in the tera-ohm range. To stop the amplifier from leaving the correct operating point due to leakage currents, a reset pulse is applied to the amplifier every few seconds. During this pulse the MOSFET in the feedback path is transferred to the linear region. Thereby, the feedback resistance is lowered significantly and the correct input voltage is reestablished.

Additionally, materials with a very high specific electrical resistance are used for all parts with contact to the input of the amplifier. The input stage of the amplifier consists of MOSFETs with a differential input capacitance of $C_{\text{in,diff}} = 3.75\,\text{pF}$. The feedback capacitors have a value of $C_f = 100\,\text{fF}$. In the laboratory test the amplifier offers a very low equivalent noise charge of $167\,e^-$ (0.028 fC) in the frequency range from 10 Hz to 300 kHz for a differential detector capacitance of 4.7 pF, which is already heavily affected by dielectric loss noise of the PCB and the detector capacitance. In the accelerator setup the amplifier noise is dominated by interferences that mechanically couple into the detector electrodes, as visible in Figure 7.

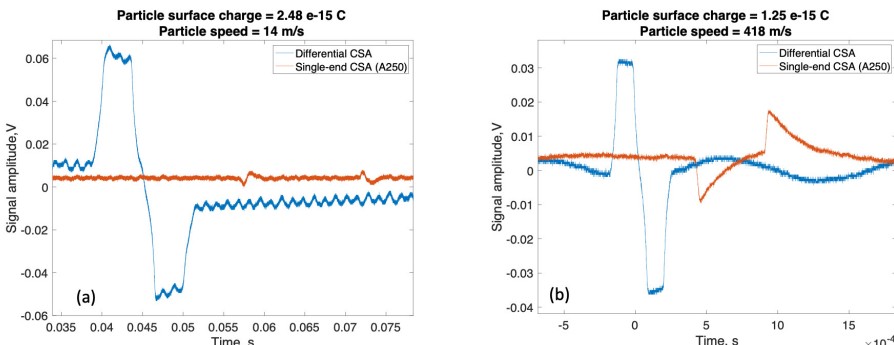

**Figure 7.** Typical inducted signals recorded two types of dust detectors: (**a**) one low-speed particle (14 m/s) and (**b**) one high-speed particle (418 m/s).

The potential of the electrode is unbalanced by a negative charge of the same amount as the positive charge on the accelerated dust particle. The difference to the voltage amplification of a charge sensitive amplification is the capacitance $C_f$ in addition to a

resistor $R_f$. A better sensitivity could be achieved with a matched-filter-based evaluation that considers both the positive and negative part of the signal. For the simple case of a positive-only evaluation, the particle will induce a charge in one of the electrodes, resulting in a charge gain for an ideal amplifier of:

$$A_{\text{q,diff}} = \frac{1}{C_f} \tag{4}$$

For the measurements, only one output of the differential CSA is recorded; assuming a sufficient common mode suppression, the output voltage is therefore divided by two, resulting in:

$$A_{\text{q,diff,se}} = \frac{1}{2 \cdot C_f} \tag{5}$$

Before digitizing, the signal is passed trough a simple bandpass filter and further amplified by a factor of 2.5, resulting in an overall charge gain of $A_{\text{q,tot}} = 13 \frac{\text{V}}{\text{pC}}$.

## 2.4. Particle Selection Unit

The PSU system is indispensable to the electrostatic dust accelerator and allows us to select particles of a defined speed and charge. The PSU system normally monitoring the beam in our 2 MV dust accelerator was integrated and tested at the small accelerator, as shown in Figure 8. The PSU system has three single-ended dust detectors, one deflection stage and one control unit. The amplified induction signals of three beam detectors provide the input signals for the PSU, which processes the singles after the analog-digital converters (ADC). Digital filters are used to recognize the signal down to 0.1 fC. The control unit calculates the speed, charge and mass of all accelerated particles and triggers the deflection plates in order to select particles within a known speed and mass range. All unselected particles are deflected out of the beam and hit the inner wall of the the vacuum pipes. The control unit provides further speed-dependent trigger pulses at variable distances along the beam line. Such trigger signals can be used to start the data recording in the experiment chambers.

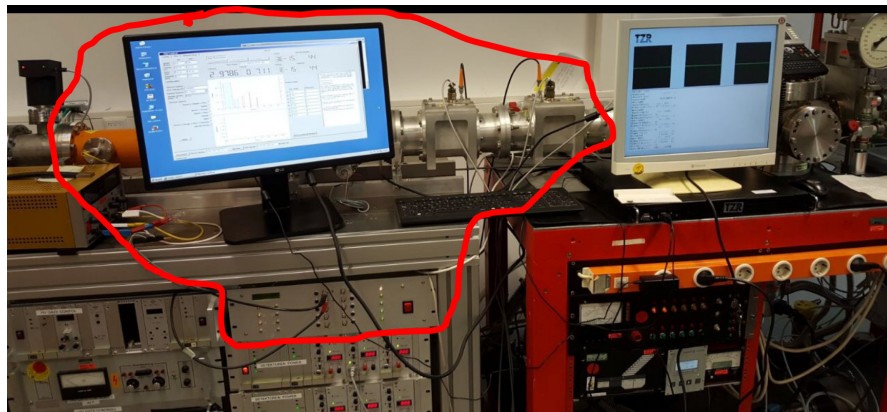

**Figure 8.** Small dust accelerator with the PSU system (labeled in red line). The deflection plates are contained in the yellow vacuum pipe behind the PSU monitor.

## 2.5. Post-Stage Linac

According to Equations (1) and (2), the final particle speed launched from the dust source depends on the particle size, the material and the potential of the needle electrode. The dust speed can be increased by increasing the voltage connected to the dust source. Unfortunately, intense high voltages at the needle can cause field emission and they can destroy the sharp tip [4,20]. A later acceleration stage is necessary to obtain more high-speed particles. Therefore, we developed a compact linac (see Figure 9) with a 600 mm

long vacuum pipe with standard CF 100 flange interfaces. The linac has 6 drift tubes with an inner diameter of 40 mm.

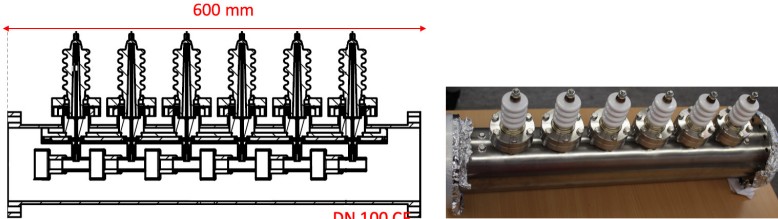

**Figure 9.** Schematic diagram and photo of the vacuum pipe and high-voltage feedthroughs of the post-stage linac.

When a particle passes the differential detectors, two signals are generated by the differential detector and sampled with a field-programmable gate array (FPGA) to calculate the particle speed. Taking the accelerating voltages of the dust source and the linac into account, the FPGA calculates the pulse train necessary to switch the polarity of the drift tubes as a particle passes through in real time. A high-voltage circuit constantly charges the drift tubes to −20 kV. When a positively charged particle approaches a drift tube, it is attracted and therefore accelerated. When it reaches the center of the drift tube—which acts like a Faraday shield—and the FPGA signal arrives, the tube is grounded by a high-voltage MOSFET switch. In this way, the particle sees an accelerating field again as it leaves the tube. An initial setup, with a 16 kV dust source voltage and 16 kV in 5 linear acceleration stages, increased the final speeds of particles by a factor of about 2.23 (see Figure 10). Due to electromagnetic compatibility (EMC) problems, the high voltage supply is currently being rebuilt.

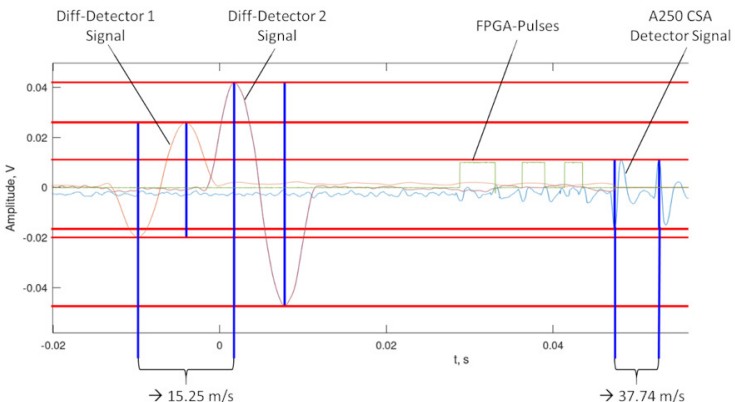

**Figure 10.** Preliminary test results of the linac stage. The particle speed is measured by the flight time between the first and third tube in the differential detector (15.25 m/s) and a single end detector after the linac stage (37.47 m/s).

## 3. Results

Various conductive metal and coated mineral powders were accelerated in the upgraded small dust accelerator. Table 4 shows a selection of the launched dust particles and their parameters. Table A1 shows the compositions of iron and copper particles provided by the supplier (ThermoFisher). Permanent particle beams were observed with iron, copper and carbon. Figure 11 shows the speed-mass, charge-mass and size distributions of 20,081 registered copper particles. A copper particle with a diameter of 26 nm (m = $6.27 \times 10^{-19}$ kg) has a final speed of 6.7 km/s. According to Equations (1) and (2), the smallest particles obtain the highest speeds, but also the lowest charges. Due to their tiny surface charges, the measurement of fast particles nanometers in diameter is very tricky

even with our state-of-the-art differential detector. A more sensitive PSU system specially designed for the differential detector is under development.

**Table 4.** Parameters of accelerated dust particles at the small dust accelerator.

| Material | Supplier | $U_{cc}$, kV | Event Number | Grain Size, μm | Velocity, m/s |
|---|---|---|---|---|---|
| Iron | ThermoFisher | 7–20 | 16,218 | 0.02–10 | 11–7140 |
| Copper | ThermoFisher | 9–12 | 20,081 | 0.02–10 | 12–6667 |
| Carbon | ThermoFisher | 3–12 | 2570 | 0.4–12 | 10–1530 |
| SiO$_2$ [1] | Dr. V. Steck [2] | 3–9 | 810 | 0.4–0.9 | 10–1470 |
| Peridot [3] | Dr. J. Hiller [2] | 15 | 761 | 0.7–3 | 17–939 |

[1] Polypyrrole-coated. [2] Personal communication. [3] Lead-coated.

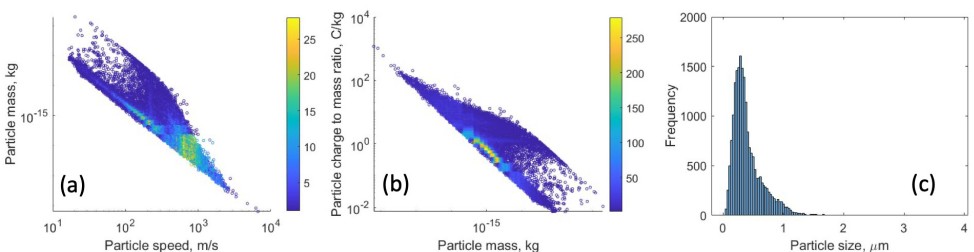

**Figure 11.** Parameters of accelerated copper particles with a voltage of 19 kV. (**a**) Particle speed and mass distribution; (**b**) Particle charge-to-mass ratio varies with mass; (**c**) Particle size distribution.

We observed single particles of polypyrrole-coated SiO$_2$ and lead-coated peridot. However, the acceleration process of these coated particles usually broke down in a short operation time. Most of the filled dust particles in the reservoir were adhered on the needle electrode, as shown in Figure 12. It is not yet fully understood why particles adhere on the needle. More detailed investigations of the coated particle parameters (such as particle geometries, coating materials, thickness, etc.) are still required.

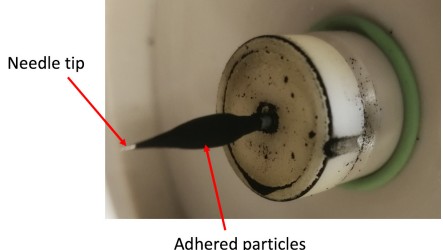

**Figure 12.** Needle electrode with adhered dust particles.

## 4. Discussion

We upgraded our small dust accelerator with one newly developed dust source, one differential dust detector and one linac stage. The PSU system was also tested in the new beam line. Various conductive metal and coated powders were tested. The accelerator has already been used for a wide variety of experiments to investigate the interaction of the high-speed dust particles with a solid surface and thin foil, such as impact cratering/penetration, impact fragmentation, impact ionization, impact flash, etc. Several calibration tests of the in situ dust detector have also been performed with this small dust accelerator. A new PSU system with differential detectors is under development. Improvements in the acceleration of a wider variety of dust materials and detection and selection of dust particles has led to an increasing number of applications.

**Author Contributions:** Conceptualization, Y.L.; Methodology, Y.L., M.B., S.K., H.S., J.S., C.M., M.S. and A.M.; writing—original draft preparation, Y.L., M.B. and S.K.; writing—review and editing, Y.L., M.B., S.K. and R.S.; supervision, Y.L. and R.S. All authors have read and agreed to the published version of the manuscript.

**Funding:** This work was partially supported by Deutsche Forschungsgemeinschaft (DFG) grant BE 2256/31-1 and grant SR 77/6-1.

**Institutional Review Board Statement:** Not applicable.

**Informed Consent Statement:** Not applicable.

**Data Availability Statement:** The data that support the findings of this paper are available on request from the corresponding author.

**Acknowledgments:** The authors wish to thank Manfred Hartling and Sebastian Bugel for their technical support in hardware development and accelerator operation.

**Conflicts of Interest:** The authors declare no conflict of interest.

## Abbreviations

The following abbreviations are used in this manuscript:

| | |
|---|---|
| ADC | Analog–digital converters |
| ASIC | Application-Specific Integrated Circuit |
| CSA | Charge-sensitive amplifier |
| EMC | Electromagnetic compatibility |
| FPGA | Field-programmable gate array |
| MOSFET | Metal-Oxide-Semiconductor Field-effect Transitor |
| PSU | Particle-selection unit |
| PTFE | Polytetrafluorethylen |
| TTL | Transistor–Transistor Logic |

## Appendix A

**Table A1.** Dust Compositions.

| Material | Supplier | Shape | Composition |
|---|---|---|---|
| Iron | ThermoFisher | Spherical | Iron 98.19%, Carbon 0.68%, Oxygen 0.46% and Nitrogen 0.67% |
| Copper | ThermoFisher | Spherical | Copper 99.9%, Sliver < 10 ppm, Al < 10 ppm, Carbon 19 ppm, Iron < 10 ppm, Nickel < 10 ppm, O2 4650 ppm, Lead < 20 ppm, Silicon < 20 ppm, Tin < 20 ppm and Zinc < 20 ppm |

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
