# Peer review of "Upgrades of a Small Electrostatic Dust Accelerator at the University of Stuttgart"

_applsci, doi:10.3390/app13074441_

Round 1

Reviewer 1 Report

This paper investigates newly developed dust source, focusing lens, differential detector and linac stage were successfully installed and tested in the beam-line. It is interesting. However, The description of the paper is relatively rough. For example, the composition, size distribution of dust are not explained in the paper. 

The paper mentioned the dust in the lunar exploration mission. I believe that the speed of cosmic dust is often greater than 10km/s, while the speed of dust particles simulated in this paper is 418m/s, 7140m/s(see figure 6), which is lower. So how to ensure that the laboratory research is more practical and help us understand and solve practical scientific problems.

Author Response

A(1): Table A1 was added to show particle compositions. Figure 11 (c) shows particle size distribution. 

A(2) : The dust environment above the lunar surface contains cosmic dust particles, the ejecta particles and levitated particles. The major of ejecta particles generated by the bombardments of cosmic dust would be launched from the lunar surface with speeds up to several 100 m/s. The levitated dust particles are charged and launched with speeds up to tens m/s by local electric fields. The upgraded small accelerator can be used to calibrated and test dust detectors for the latter two populations.

Reviewer 2 Report

In this paper, the authors describe the upgrade of their dust accelerator. The details of the upgrade for different subsystem are well introduced, and some testing results are illustrated and analyzed preliminarily. The presentation and English language are good. However, the readers may be interested in the comparison before and after the upgrade. I suggest the authors give some tables or figures related to the comparison.

Author Response

A: Table 1 was added to show the comparison between previous setup and upgrades setup

Reviewer 3 Report

The paper presents the upgrade of an electrostatic dust accelerator. The dust source, focusing lens, differential detector and linac stage were newly developed, and the paper describes the detail of the instrumentations and tests. The system demonstrates acceleration of micron and sub micron-sized metal particles and coated mineral materials to the velocity between 10 m/s and 7km/s. 

The paper describes the system enough for publication, I suggest only a minor comment as follows.

- line 14 in p6  

... four short cylindrical tubes esch 5cm ...

-> ... four short cylindrical tubes each 5cm ...

- section 1, page 2
The paper presents the upgrades of a dust accelerator from the previous design. Please review characteristics, obtained results, and remained problems in the previous design. This will enhance the importance and rationales of the upgrades presented in the current manuscript.

- section 2.5, page 8
I suggest that the test setup is described in a separated subsection, not in "2.5 Post-stage linac". 

Author Response

A1: Errors were corrected

A2: Table 1 was added to show the comparison between previous setup and upgrades setup

A3: The test setup was moved to first paragraph of Section 2, and was described together with the previous setup and the PSU setup.

Reviewer 4 Report

I guess the paper will be interesting for many readers. At least it was interesting for me to read your paper and to expand my knowledge of non-conventional accelerators. It turned out dust accelerators got substantial progress in recent years. I would recommend to revise only one thing. The discussion section should be substituted by the Conclusion or optionally the Conclusion must be added as the next section after Discussion. In the Conclusion you must clearly summarize what particular improvements you have made and what new parameters and opportunities you have obtained due to these upgrades. Good luck!

Author Response

A: As required in the guide of manuscript preparation. MDPI suggested to describe all results and improvements in ‘Discussion’ section. The ‘Conclusions’ section is required only if the discussion is unusually long. Hence, we summarized all results, finding and future application of the instrument briefly in discussion.

Round 2

Reviewer 2 Report

The authors added the comparison between previous and upgraded design by listing in a table. It’s helpful for the readers to quickly realize their differences. However, the experimental or testing comparison still lacks in the revision. How do you prove the dust accelerator have been upgraded?

Author Response

Our previous small accelerator was used to test dust sources and dust samples The upgraded version can be additionally used for tests and calibrations of dust detectors with different requirements. For example, the PSU system is suitable for tests with selected particles (See Figure 8 in the manuscript), and the linac stage is used for particles with relatively high speeds (See Figure 9 in the manuscript). The other upgrades such as the new dust source and the new dust detector are also described in the Section 2.